# Intraoral Ultrasonography for the Exploration of Periodontal Tissues: A Technological Leap for Oral Diagnosis

**DOI:** 10.3390/diagnostics14131335

**Published:** 2024-06-24

**Authors:** Matthieu Renaud, Mickael Gette, Alexis Delpierre, Samuel Calle, Franck Levassort, Frédéric Denis, Gaël Y. Rochefort

**Affiliations:** 1Faculty of Odontology, Tours University, 37000 Tours, France; mickael.gette@etu.univ-tours.fr (M.G.); alexis.delpierre@univ-tours.fr (A.D.); frederic.denis@univ-tours.fr (F.D.); gael.rochefort@univ-tours.fr (G.Y.R.); 2Department of Medicine and Bucco-Dental Surgery, Tours University Hospital, 37000 Tours, France; 3Bioengineering Biomodulation and Imaging of the Orofacial Sphere, 2Bios, Odontology Department, Tours University, 37000 Tours, France; 4N2C U1069 INSERM, Tours University, 37000 Tours, France; 5GREMAN, Université de Tours, CNRS, INSA-CVL, 26 Rue Pierre et Marie Curie, 37100 Tours, France; samuel.calle@univ-tours.fr (S.C.); franck.levassort@univ-tours.fr (F.L.); 6EA 75-05 Education, Ethique, Santé, Faculté de Médecine, Université François-Rabelais, 37000 Tours, France; 7iBrain U1253 INSEM, Tours University, 37000 Tours, France

**Keywords:** ultrasonography, periodontal tissue, periodontology, high-frequency ultrasound imaging, periodontal imaging

## Abstract

Introduction: Periodontal disease is an infectious syndrome presenting inflammatory aspects. Radiographic evaluation is an essential complement to clinical assessment but has limitations such as the impossibility of assessing tissue inflammation. It seems essential to consider new exploration methods in clinical practice. Ultrasound of periodontal tissues could make it possible to visualize periodontal structures and detect periodontal diseases (periodontal pocket measurement and the presence of intra-tissue inflammation). Clinical Innovation Report: An ultrasound probe has been specially developed to explore periodontal tissues. The objective of this clinical innovation report is to present this device and expose its potential. Discussion: Various immediate advantages favor using ultrasound: no pain, no bleeding, faster execution time, and an image recording that can be replayed without having to probe the patient again. Ultrasound measurements of pocket depth appear to be as reliable and reproducible as those obtained by manual probing, as do tissue thickness measurements and the detection of intra-tissue inflammation. Conclusions: Ultrasound seems to have a broad spectrum of indications. Given the major advances offered by ultrasound imaging as a complementary aid to diagnosis, additional studies are necessary to validate these elements and clarify the potential field of application of ultrasound imaging in dentistry.

## 1. Introduction

The periodontium, defined as all tissues supporting the dental organ, is composed of alveolar bone, root cementum, periodontal ligament, and gingival tissues. Each periodontal component has its own highly specialized structure, and these structural features directly define their function [1]. The sulcus is the space between the gingival part and the tooth and is composed of the supracrestal attachment, including the junctional epithelium and connective tissue. Violation of this epithelial-connective attachment by dental restorative material or bacterial proliferation within biofilm is associated with inflammation of periodontal tissues. This is followed by loss of connection and permanent resorption of alveolar bone, defining periodontal disease [1].

Periodontal disease is an infectious syndrome presenting inflammatory aspects with a multifactorial etiology [2]. This disease can be divided into two subcategories: inflammatory lesion of the periodontal system, without bone loss (gingivitis) or with bone lesions (periodontitis) [3]. While all of these diseases affect a significant proportion of the French population, to varying degrees, they are responsible for more than 45% of severe attachment loss (i.e., >5 mm) and 30 to 40% of total tooth extractions [4,5]. Furthermore, they remain insufficiently detected and treated. Similar prevalence can be found in other European countries and more generally throughout the world [6].

Periodontal disease most often depends on a specific biofilm. If this biofilm is not regularly reduced or eliminated by brushing, we witness microbial dysbiosis, an ecological modification favoring greater pathogenicity. In this context, hematogenous dissemination of bacteria and inflammatory mediators is triggered, leading to systemic inflammation in some individuals. In addition to other biological mechanisms, the resulting pathological imprint affects multiple organs, with strong evidence of particularly damaging links with the endocrine, cardiovascular, pulmonary and neurological systems [7] Periodontitis can also have a direct impact on a number of organs and is increasingly associated with the emergence of alarming health problems such as in obesity, where a co-influence between the bacterial flora of the patients’ periodontium and their level of overweight appears to be linked [8]. There are also links between polycystic ovary syndrome and periodontal diseases, where a downward spiral of inflammation and pathogen development appears to be self-reinforcing [9]. There is now strong evidence that good management of periodontitis helps control glycemia in patients suffering from type 2 diabetes, demonstrating once again the impact that periodontal health can have on systemic pathologies [10].

Radiographic evaluation is an essential complement to the clinical assessment of the periodontium. Radiographs of normal periodontium include an intact lamina dura, no signs of bone loss in furcation areas, and a distance of 2 mm between the most coronal portion of the alveolar bone crest and the cemento-enamel junction of the tooth [11]. Retro-alveolar radiographs are indicated for the evaluation of horizontal and vertical bone lesions. They provide information about the tooth and surrounding alveolar bone. But conventional radiography has the drawback of using ionizing radiation technology, and even though the latest generation of devices tends to reduce exposure to these ionizing rays, they will always be present, with all the risks that may entail [12].

Additionally, it is impossible to assess tissue inflammation using this type of imaging. Only the presence of bone demineralization can be detected, which is a late sign of bone inflammatory reaction. The retro-alveolar image can only show bone loss occurring several months after the onset of epithelial attachment loss and the inflammatory reaction of the periodontal tissues. Relying solely on retro-alveolar radiography does not allow early diagnosis of periodontal disease. Although radiographic examination is currently the most widely used complementary examination in clinical practice, we have highlighted some of its limitations [13]. Given these limitations, it seems essential to consider new means of exploration in clinical practice.

As recent data in the literature show, ultrasound of periodontal tissues could make it possible to visualize periodontal structures and detect periodontal diseases (periodontal pocket measurement and the presence of intra-tissue inflammation) [14]. Directly measuring the periodontal pocket and detecting periodontal tissue inflammation on ultrasound images could facilitate the early diagnosis and treatment of periodontal disease. In addition, the imaging of periodontal tissues and their live analysis without the use of ionizing radiation constitute a major advance compared to routine retro-alveolar radiography. However, periodontal ultrasonography is not intended to replace all other means of evaluating periodontal tissues. But the associated prospects are immense, with numerous applications ranging from periodontal assessment to therapeutic re-evaluation and surgery [15]. Thus, ultrasound offers the prospect of a paradigm shift in the diagnosis and monitoring of periodontal disease, by being more reproducible and efficient.

In this context, we were working with our teams on the development of an intraoral ultrasound probe (research consortium GREMAN UMR 7347, CHU de Tours, and U1253 IBrain and Carestream Dental©). This device is specially dedicated to the exploration of periodontal tissues and to obtaining quality live images of the human periodontium.

The objective of this clinical innovation report is to present this new innovative device and to expose its potential as an ultrasound imaging tool for the exploration of periodontal tissues and the visualization of periodontal anatomical structures. Furthermore, the secondary objective of this article is to propose a clinical and technical user guide for this ultrasound probe and to define its advantages and disadvantages.

## 2. Clinical Innovation Report

### 2.1. Ultrasound Device

The device used for this study is currently being developed by the Carestream Dental company in clinical collaboration with the Odontology Department at Tours University Hospital. It consists of a portable and wireless device, designed to be able to image and analyze the soft tissues of the oral cavity (Figure 1). Based on a high-frequency transducer (approximately 20 MHz), that system allows acquiring high-resolution images (approximately 100 µm axial and lateral resolutions) of the soft tissues of the oral cavity. The ultrasonic flow can be visualized on a paired smartphone or tablet hosting the mobile application.

### 2.2. Protocol of Clinical Examination

To use the ultrasound imaging device with the intraoral probe, the patient is seated comfortably in a dental chair, in a semi-sitting position. The head of the ultrasonic device is smaller than intraoral cameras used for optical imaging, making it easier to cover the entire oral cavity. The probe head is protected by a disposable cover. The patient’s mouth is half-open to allow tension-free access to the tissues. The probe is delicately presented in the mouth by the operator. A coupling gel is used between the probe head and the tissues to enable the transmission of ultrasounds between the emitting source and the tissues (Figure 2). The probe head is positioned orthogonally and parallel to the tissues to be explored. For this purpose, the long axis of the tooth is used as a reference point, parallel to the axis of the probe head. The reading zone of the probe is positioned as orthogonally as possible to the surface of the soft tissues, to avoid any form of deformation (Figure 3).

For ease of use, scanning begins with the vestibular surfaces of the upper and lower jaws, moving back and forth and right to left (from tooth 17 to 27, then from tooth 47 to 37), with a back-and-forth movement lasting a few seconds per tissue surface, equivalent to about 2–3 s to scan a tooth between its two proximal edges. Next, the probe is applied in the same manner to the lingual surfaces, following the same scanning protocol.

Since it is connected to the computer via Wi-Fi, images can be read live, for better use of the system and more accurate image capture. The images can also be saved and organized according to the traditional dental schema, for specific analyses and measurements of anatomical elements either directly or extemporaneously.

### 2.3. Ultrasound Imagery of Periodontal Tissue

Figure 4 shows healthy periodontal structures created by the ultrasound probe. All periodontal structures (hard and soft tissues) can be observed. Figure 4A shows enamel, cementum, and the upper part of the alveolar bone. The ultrasound signal highlights these densely packed structures, and the echo allows radiopaque areas to appear in the image. The gingival epithelium is keratinized, revealing a structure that is also radiopaque on the surface, then more radiolucent in the depth of the tissue. In addition, the connective tissue part also appears more radiopaque, given its high composition of different fiber types. Figure 4B shows a merge of the schematized anatomical structures and ultrasound image for simplified didactic reading. This superimposition shows the easy readability of periodontal histological structures by ultrasound imaging. Figure 4C shows the position of the enamel-cement junction and the epithelial-conjunctival attachment zone. The epithelial and conjunctival attachments are easily observable, as is the depth of the sulcus.

Figure 5 shows periodontal structures in pathological situations. In Figure 5A, the green arrows show the presence of enamel structure in the absence of deposits on its surface, while in Figure 5B, the green arrows show the presence of calculus. Indeed, the ultrasound signal appears less dense and the structure thicker due to a more diffuse ultrasound signal. In Figure 5C, the green arrows indicate the presence of a physiological epithelial-conjunctive attachment. These structures are clearly visible and well-defined. The sulcus space is measurable. Otherwise, Figure 5D indicates the presence of intratissular inflammation. The ultrasound signal is diffuse, with radiolucent areas showing the presence of water in the tissue. Anatomical structures are no longer clearly visible or determined.

### 2.4. Interests

The use of ultrasound imaging in periodontology offers numerous clinical advantages. Above all, ultrasound does not use an X-ray source, making it safe, non-invasive and unrestricted. Using an intraoral ultrasound probe is also painless for the patient. In addition, this type of imaging allows direct and live reading of the tissue portion observed, allowing better clinical discrimination [16]. Visualization of soft tissues and all periodontal structures is easily possible, thanks to training in both operating the probe and reading images, as is the practice in all fields where this type of imaging is used [17]. With this type of ultrasound imaging, we can objectify tissues such as enamel, the enamel-cement junction, the external part of the alveolar bone, the sulcus area, the part of the free gingiva, the epithelial attachment, and the connective attachment [18]. Also note that periodontal pocket measurement is possible and easy thanks to digital measurement without the use of a periodontal probe [19]. Gum tissue thickness can also be measured in the same way, providing additional clinical information. The presence of calculus can also be detected on the root surface. This possibility of detection is an advantage to reduce pocket measurement bias.

Ultrasound waves used in ultrasound imaging propagate in media with different acoustic impedances. As a result, a local deformation is transmitted via a mechanical wave which propagates step by step through these media. Propagation media have their own characteristics, with different repercussions on waves with intrinsic elastic properties [20,21,22]. This makes it possible to visualize the presence of water in the tissues, which reflects the inflammatory phase. Thus, the detection and visualization of an intra-tissue inflammatory zone are made possible by this live imaging system. Likewise, the presence of subgingival calculus is also visible on this type of imaging.

On the other hand, ultrasound probes require the use of a coupling gel between the probe head and the tissue analyzed, which makes handling the probe slightly delicate despite the safety of the gel.

## 3. Discussion

Periodontal probing, even essential for monitoring the extent of periodontal disease, remains an uncomfortable procedure for the patient, requiring a good level of training to achieve acceptable intra- and inter-operator repeatability and reproducibility [23].

Two techniques are currently available on the market: manual probing with a graduated periodontal probe or an electronic probe. In both cases, the dentist inserts the tip of the instrument into the sulcus to determine the depth of the periodontal pocket, causing pain and bleeding for the patient. For both techniques, the time required by a qualified dentist to collect all the data is on average ten minutes [24].

Here we can see the immediate advantages of using ultrasonography to study the periodontium: no pain, no bleeding, faster completion time, and image recording that can be replayed without having to probe the patient again.

Some ultrasound probes have already been presented in the literature in the past. These probes had the major disadvantage of not being adapted to the oral cavity due to the lack of miniaturization and the presence of a suitable transducer making it possible to produce good-quality images [15]. This technological barrier was removed thanks to the work and development of this new probe by a research consortium (GREMAN UMR 7347, CHU de Tours, U1253 IBrain, and Carestream Dental©). This development made it possible to create a device suitable for intraoral manipulation with a miniaturized transducer and allowing the production of images of sufficient resolution. In addition, this research made it possible to develop an ultrasound probe with low production costs allowing efficient use of the device in daily practice. It is also noted that the evolution of electronic technology has made it possible to lighten the weight of the probe and eliminate the wire, allowing a wireless connection of the device.

High-resolution images are as accurate as or more precise than conventional imaging. Non-ionizing imaging eliminates the discomfort of periodontal probing and retro-alveolar imaging. It also provides a more reliable reading of areas that are difficult to access clinically and difficult to measure with traditional periodontal probing. Ultrasound measurements of pocket depth appear to be as reliable and reproducible as those obtained by manual probing, even with an electronic probe [15]. The same appears to be true for tissue thickness measurements and the detection of intra-tissue inflammation. However, more in-depth clinical studies should be able to definitively verify and validate these hypotheses. The visualization and analysis of periodontal soft tissues appear to be a revolution in the overall clinical and paraclinical approach. Ultrasound imaging can provide additional clinical information, such as tissue thickness, quality of keratinized tissue, presence of inflammatory infiltrates, or interdental areas with infrabony defects [25]. In the case of interdental areas, especially in the presence of infrabony defects, ultrasound imaging can provide additional information to clarify the condition of these areas, such as the position of the periodontal attachment, the presence of intra-tissue inflammation, and the loss of density of the interdental septum. Studies are needed to clarify these elements.

Elastographic analysis can even measure the elasticity of periodontal tissues [26]. Doppler ultrasound, for its part, makes it possible to visualize and measure vascularization, a predictive factor for gingival surgery [27]. Ultrasound imaging is also an important way to increase the sensitivity and specificity of our diagnosis.

Ultrasound images could also open new perspectives for early diagnosis, such as the distinction of the sulcular space and measurement between the cemento-enamel junction and sulcus [28]. With standardized measurements of this type, the process could even be automated, and the data regularly integrated into periodontal charts, providing new details for more accurate diagnosis. We would then have a reliable and less time-consuming system for monitoring patients.

At present, ultrasound examinations take around 30 min per patient, and reading the images generated by our device requires time for learning and acculturation. This is why the replacement of manual periodontal measurements by an ultrasound device seems to be moderate. Despite the advantages provided by the device, evaluation with a probe remains faster to date and relatively non-invasive. Training in reading images would allow rapid acclimation to the use of ultrasound in current practice to improve the efficiency of the device.

## 4. Conclusions

Ultrasound appears to have a wide range of indications, from initial periodontal examination to post-treatment follow-up of periodontal diseases, from periodontal surgery to oral dermatology, with early detection of hyper-inflammatory lesions characteristic of potentially malignant lesions. However, more research is needed to validate the possibility of using ultrasound as a diagnostic tool: detection of inflammation, level and quantity of inflammation, tissue thickness, measurement of periodontal pockets, presence of bony defects or demineralization. Ultrasound seems to be, at present, a complementary diagnostic tool to routine clinical examination.

There are also numerous benefits for patients, including less pain, less or even no bleeding, and shorter recording times.

From a scientific and medicolegal viewpoint, recording allows images to be replayed, more accurate monitoring over time, simplified training for dentists wishing to start up in this field and, finally, formal proof of pathology development level. For now, technical expertise is needed to use this ultrasonography.

Given the major advances offered by ultrasound imaging as a complementary aid to diagnosis, can we today speak of calling into question a certain number of gold standards? Additional studies are needed to validate these elements and clarify the potential field of application of ultrasound imaging in dentistry.

## Figures and Tables

**Figure 1 diagnostics-14-01335-f001:**
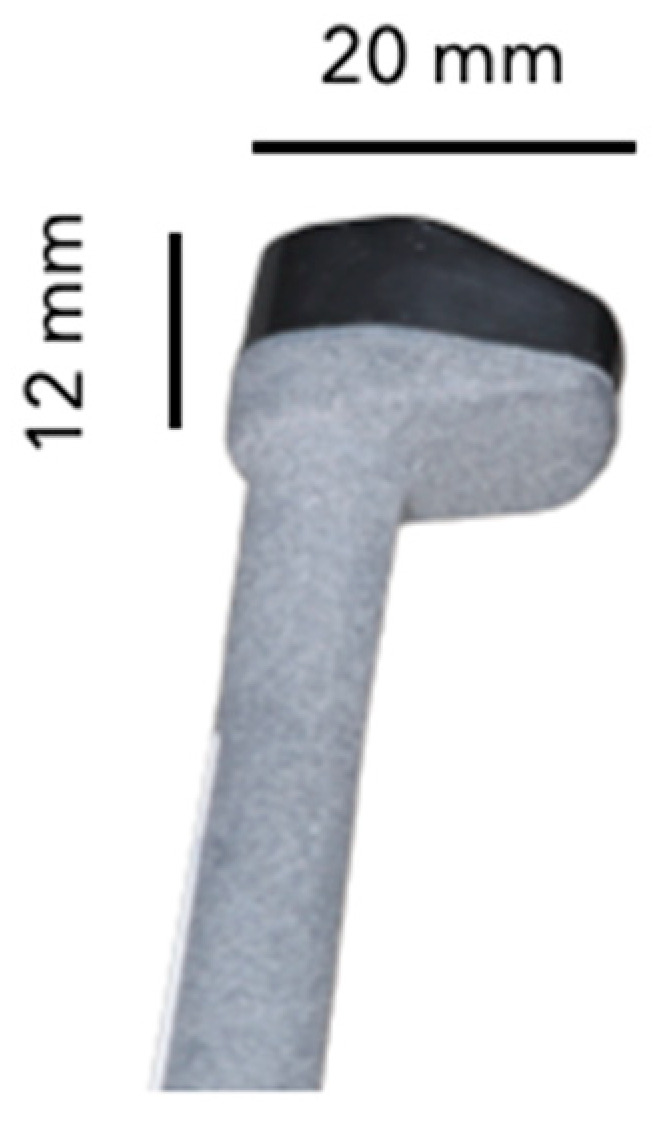
Ultrasound device: the head of the device.

**Figure 2 diagnostics-14-01335-f002:**
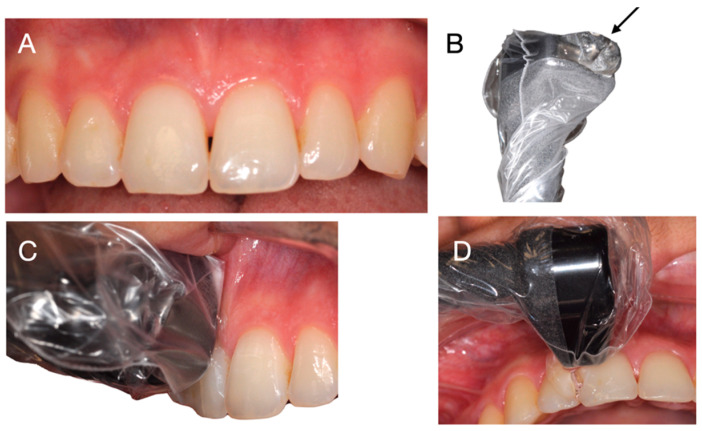
Clinical utilization of the device. (**A**) Clinical aspect. (**B**) Head of the device with coupling gel indicated by a black arrow. (**C**) Lateral view: position of the head of the device on the tooth and tissues. (**D**) Frontal view: position of the head of the device on the tooth and tissues.

**Figure 3 diagnostics-14-01335-f003:**
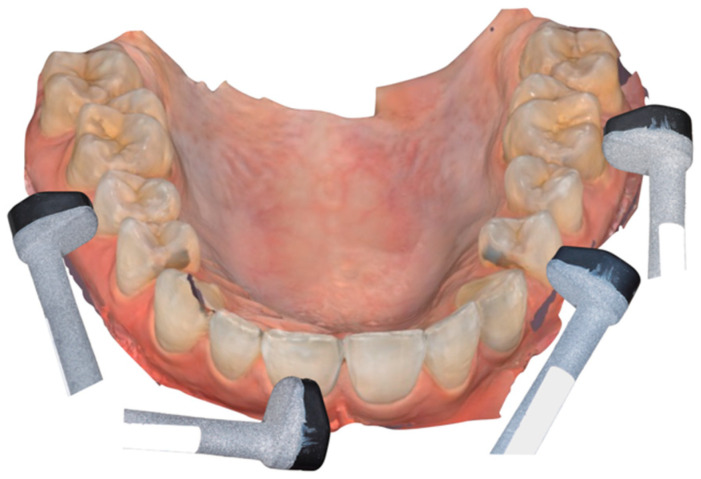
Different positions of the device in function of the clinical situation. The probe head is positioned orthogonally and parallel to the tissues to be explored.

**Figure 4 diagnostics-14-01335-f004:**
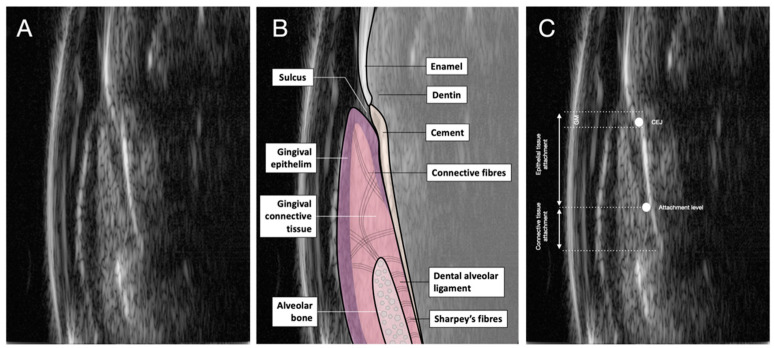
Ultrasound image of the periodontium. (**A**) View of the different structures of a healthy periodontium. (**B**) Merged with histological aspects. (**C**) Different structures of the periodontium with the location of the cementum-enamel junction, attachment level, and connective and epithelial tissue attachments.

**Figure 5 diagnostics-14-01335-f005:**
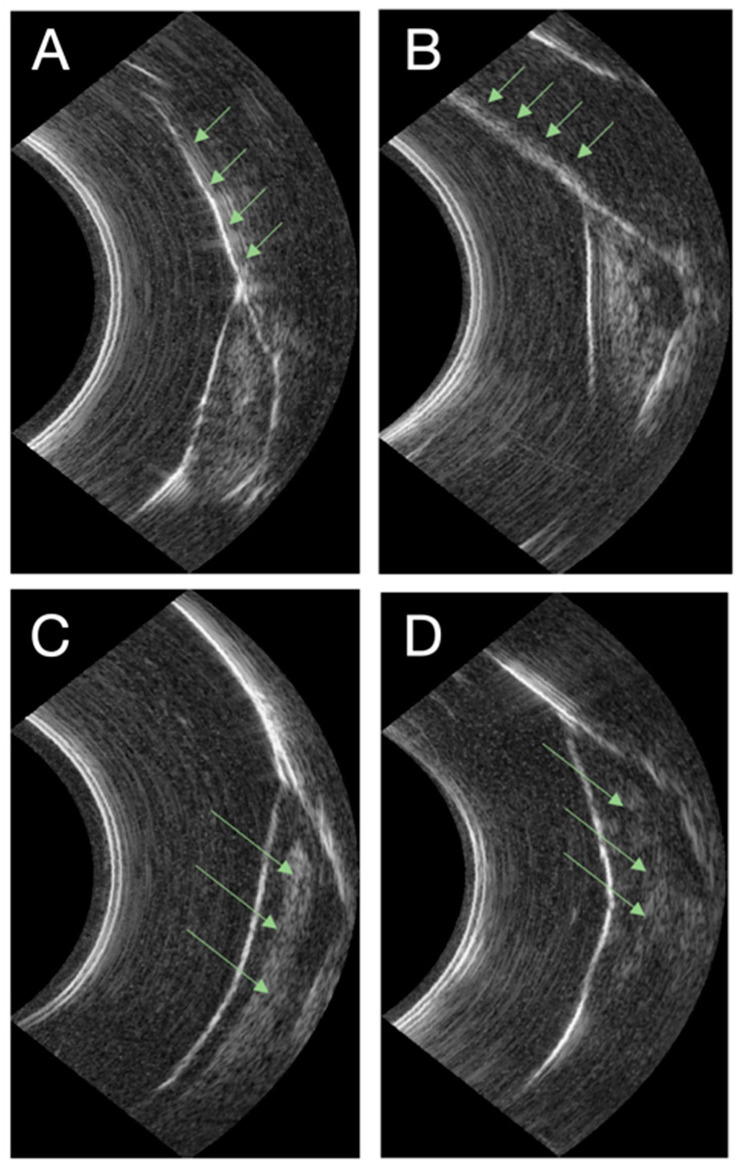
Ultrasound image of pathological periodontium. (**A**) Green arrows indicate an enamel without the presence of tartar. (**B**) Green arrows indicate enamel with the presence of tartar (excess of thickness). (**C**) Green arrows indicate healthy epithelial-connective tissue. (**D**) Green arrows indicate the presence of inflammation in epithelial-connective tissue (diffuse ultrasound signal).

## Data Availability

No new data were created or analyzed in this study.

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
