# Peer review of "Intraoral Ultrasonography for the Exploration of Periodontal Tissues: A Technological Leap for Oral Diagnosis"

_diagnostics, 2024, doi:10.3390/diagnostics14131335_

Round 1

Reviewer 1 Report

Comments and Suggestions for Authors
  1. The manuscript is well written and needs a recent development in diagnosis.
  2. This study introduces a noninvasive method for the diagnosis of periodontal diseases and forms high-resolution images due to the development of this new ultrasound probe by a research consortium (Greman UMR 7347, 220 CHU de Tours, U1253 IBrain and Carestream Dental©).
  3. Technical expertise is needed to use this ultrasonography.
  4. It lacks extensive clinical trial data to substantiate the claims. More research is needed to conclude the final diagnosis as it has not been used in various types of periodontal disease till now.

Author Response

Response: I would like to thank you about your kind return about this technical note. Elements were added in the conclusion to temper the comments on the subject of ultrasound. Clinical validation studies are indeed necessary to shed light on the possibility of making a diagnosis using ultrasound.

However, more research is needed to validate the possibility of using ultrasound like a diagnostic tool: detection of inflammation, level and quantity of inflammation, tissue thickness, measure of periodontal pocket, presence of bony defect or demineralization. Ultrasounds seems to be, at these days, a complementary diagnosis tool to a routine clinal examination.”

“From a scientific and medico-legal viewpoint, recording allows images to be replayed, more accurate monitoring over time, simplified training for dentists wishing to start up in this field and, finally, formal proof of pathology development level. For now, technical expertise is needed to use this ultrasonography.”

Reviewer 2 Report

Comments and Suggestions for Authors

Review of „Intraoral ultrasonography for periodontal tissue exploration: a technological leap for oral diagnosis”

The authors present in their paper a new ultrasonic recording head, which seems to have due to reduction of the head-size advantages for oral recording. This is an interesting new development.

The article shows that you can produce ultrasonic images of periodontal tissues, but it does not compare the assessment to manual periodontal probing. I think such a comparison might improve the study.

In periodontal disease the approximal areas between the teeth often show the most advanced destruction of periodontal attachment. I think it would be of interest for the reader and for the suitability of ultrasonic periodontal assessment to show in the article how reliable ultrasonic imaging works in approximal areas especially as in the interdental areas infrabony defects are often found. The authors should perhaps add in the discussion the ultrasonic assessment of intrabony defects.

If calculus detection on the root surface is possible, this would be an advantage. It might improve the article to include some additional images of calculus detection and discuss how these images can be evaluated reliably.

I am not sure about the replacement of manual periodontal measurements with an ultrasonic device, as assessment with a probe is pretty fast and manual probing is a relatively un-invasive procedure. This should also be addressed in the discussion part.

Author Response

The authors present in their paper a new ultrasonic recording head, which seems to have due to reduction of the head-size advantages for oral recording. This is an interesting new development.

 The article shows that you can produce ultrasonic images of periodontal tissues, but it does not compare the assessment to manual periodontal probing. I think such a comparison might improve the study.

Response: I would like to thank you about your kind return about this technical note. Indeed, we wanted, firstly, to present in this article the device as well as the advantages of this system as a diagnostic tool. There are many of them and they deserve to be exhibited. A clinical study is underway in our department regarding the correlation of periodontal probing / measurement of periodontal pockets using ultrasound scans. The results will be published very soon.

In periodontal disease the approximal areas between the teeth often show the most advanced destruction of periodontal attachment. I think it would be of interest for the reader and for the suitability of ultrasonic periodontal assessment to show in the article how reliable ultrasonic imaging works in approximal areas especially as in the interdental areas infrabony defects are often found. The authors should perhaps add in the discussion the ultrasonic assessment of intrabony defects.

Response: I would like to thank you about this observation. This point was added to the discussion.

Ultrasound imaging can provide additional clinical information, such as tissue thickness, quality of keratinized tissue, presence of inflammatory infiltrates, or interdental areas infrabony defects. (25) In the case of interdental areas, especially as in the interdental areas infrabony defects, ultrasound imaging can provide additional information to clarify the condition of these areas like the position of the periodontal attachment, the presence of intra-tissue inflammation, the loss of density of the interdental septum. Studies are needed to clarify these elements.”

If calculus detection on the root surface is possible, this would be an advantage. It might improve the article to include some additional images of calculus detection and discuss how these images can be evaluated reliably.

Response: I would like to thank you about this observation. In fact, the detection of the calculus on the root surface is an advantage to reduce pocket measurement bias. Elements were added in the interests part.

The presence of calculus can also be detected on the root surface. This possibility of detection is an advantage to reduce pocket measurement bias.”

I am not sure about the replacement of manual periodontal measurements with an ultrasonic device, as assessment with a probe is pretty fast and manual probing is a relatively un-invasive procedure. This should also be addressed in the discussion part.

Response: I would like to thank you about this observation. Elements were added in the discussion to temper this point.

This why, the replacement of manual periodontal measurements by an ultrasound device seems to be moderate. Despite the advantages provided by the device, evaluation with a probe remains faster to date and relatively non-invasive. Training in reading images would allow rapid acclimation to the use of ultrasound in current practice to improve efficiency of the device.”